# Effect of Pyroligneous Acid on the Productivity and Nutritional Quality of Greenhouse Tomato

**DOI:** 10.3390/plants11131650

**Published:** 2022-06-22

**Authors:** Raphael Ofoe, Dengge Qin, Lokanadha R. Gunupuru, Raymond H. Thomas, Lord Abbey

**Affiliations:** 1Department of Plant, Food, and Environmental Sciences, Faculty of Agriculture, Dalhousie University, Halifax, NS B2N 5E3, Canada; Raphael.ofoe@dal.ca (R.O.); dn376773@dal.ca (D.Q.); lk811170@dal.ca (L.R.G.); 2School of Science and the Environment, Grenfell Campus, Memorial University of Newfoundland, Corner Brook, NL A2H 5G4, Canada; rthomas@grenfell.mun.ca

**Keywords:** *Solanum lycopersicum*, biostimulant, pyroligneous acid, vegetable production, post-harvest

## Abstract

Pyroligneous acid (PA) is a reddish-brown liquid obtained through the condensation of smoke formed during biochar production. PA contains bioactive compounds that can be utilized in agriculture to improve plant productivity and quality of edible parts. In this study, we investigated the biostimulatory effect of varying concentrations of PA (i.e., 0%, 0.25%, 0.5%, 1%, and 2% PA/ddH_2_O (*v*/*v*)) application on tomato (*Solanum lycopersicum* ‘Scotia’) plant growth and fruit quality under greenhouse conditions. Plants treated with 0.25% PA exhibited a significantly (*p* < 0.001) higher sub-stomatal CO_2_ concentration and a comparable leaf transpiration rate and stomatal conductance. The total number of fruits was significantly (*p* < 0.005) increased by approximately 65.6% and 34.4% following the application of 0.5% and 0.25% PA, respectively, compared to the control. The 0.5% PA enhanced the total weight of fruits by approximately 25.5%, while the 0.25% PA increased the elemental composition of the fruits. However, the highest PA concentration of 2% significantly (*p* > 0.05) reduced plant growth and yield, but significantly (*p* < 0.001) enhanced tomato fruit juice Brix, electrical conductivity, total dissolved solids, and titratable acidity. Additionally, total phenolic and flavonoid contents were significantly (*p* < 0.001) increased by the 2% PA. However, the highest carotenoid content was obtained with the 0.5% and 1% PA treatments. Additionally, PA treatment of the tomato plants resulted in a significantly (*p* < 0.001) high total ascorbate content, but reduced fruit peroxidase activity compared to the control. These indicate that PA can potentially be used as a biostimulant for a higher yield and nutritional quality of tomato.

## 1. Introduction

Tomato (*Solanum lycopersicum*) is among the most cultivated greenhouse vegetable crops worldwide [1], and is known to be a rich source of health-promoting phytochemicals including carotenoids, phenolics, flavonoids, and ascorbic acid [2]. These phytochemicals exhibit antioxidant properties, which protect cells against oxidative stress by scavenging reactive oxygen species. Its antioxidant properties are known to induce anticancer, anti-inflammatory, and chemo-preventive effects. Thus, contributing largely to the prevention of chronic diseases such as cardiovascular, cancer, atherosclerosis, and neurodegenerative disorders [2,3]. The flavor and dietary qualities of food, which strongly influence consumers preference, are usually associated with physical characteristics (e.g., chewability and texture) and chemical composition (pH, °Brix, elements, carotenoids, phenolics, and flavonoid) [4]. These properties can be influenced by growing conditions, environmental factors, and the genetic characteristics of the plant. As a result, current greenhouse producers seek alternative inputs which rely mostly on organic amendments to improve the yield and quality of tomato fruits. One such input is the use of pyroligneous acid (PA), which is a natural and environmentally friendly by-product of pyrolysis of plant biomass [5].

During pyrolysis, organic biomass is burnt at a high temperature under the presence of limited oxygen and the gaseous and smoke phase is condensed to produce a liquid smoke [6]. The condensed liquid smoke is stabilized by allowing it to stand for six months, which results in the formation of wood tar at the bottom, light oil at the top and condensed aqueous translucent PA. This aqueous translucent PA is also known as wood vinegar, bio-oil or liquid smoke [6]. PA has a smoky odor and the color may vary from light yellow to reddish-brown depending on the feedstock [7]. It is a complex mixture containing 80–90% water as a major component and over 200 water-soluble chemical compounds including nitrogen, phenolics, organic acids, sugar derivates, alcohols, and esters [6,8,9]. The chemical composition of PA mainly depends on the temperature, heating rate, feedstock, and residence time, and has been widely used in diverse areas including agriculture, food and medicine [6,10]. Evidence revealed that PA also contains a butanolide, a biologically active compound, that belongs to a new family of phytohormones known as karrikinolide or karrikins [11,12]. Interestingly, the signaling mechanism and mode of action of karrikins are analogous to that of known phytohormones [11,13,14], suggesting that PA at an appropriate concentration can positively influence plant growth and productivity. Furthermore, karrikins are thermal resistant, hydrophilic, and long lasting and can therefore remain highly potent at a wide range of concentrations. Several studies revealed that karrikins stimulate seed germination and regulate seedling photomorphogenesis by enhancing seedling sensitivity to light [11,12,15,16,17,18].

PA is commonly used as a biostimulant to improve plant growth and productivity [6]. Depending on the concentration, PA can be used as an antimicrobial agent [19,20], a herbicide [21], a soil enhancer [22], and an insect repellent [23] or promote root development [24,25] and microbial activities [26] when diluted. Recent studies reported that PA enhances seed germination rate, vegetative and reproductive growth of several plants species [6,24,25,27,28,29]. However, the concentration of PA applied to promote plant growth varied between studies. For instance, it was reported that the application of 1:500 (*v*/*v*) increased tomato yield but did not affect fruit nutritional quality, whereas according to Mungkunkamchao et al. [27], 1:800 PA enhanced the growth and yield of tomato. Similarly, soil drench with 20% PA increased the growth and yield of rockmelon (*Cucumis melo* var. *cantalupensis*) [30]. These suggest that the effectiveness of PA is dependent on its concentration, type of crop, and mode of application. Generally, the high acidity of PA necessitates its use at low concentrations for plant growth and productivity [6]. As such, an appropriate concentration can contain the right proportions of several bioactive compounds which induce beneficial effects on crop growth and quality [17]. Furthermore, phenolic compounds in PA induce high reactive oxygen species scavenging, reducing power activities and anti-lipid peroxidation capacity [8,31]. However, the chemical composition and individual chemical activities can be influenced by the pyrolytic temperature, as a high pyrolytic temperature between 311 and 550 °C was demonstrated to exhibit the strongest antioxidant activity [8]. It was amply demonstrated that a high PA concentration increases the availability of phenolics and organic acids that could adversely affect plant growth performance [32]. All these studies demonstrated the use of PA as a natural biostimulant with high efficacy for crop production but this was not extensively explored.

Accordingly, most studies on PA efficacy and use in crop production have focused on seed priming and foliar application. There is limited information on the efficacy of drench application on crop yield and especially on crop quality [6]. Additionally, agricultural use of PA in Canada and globally is in its infant stage due to limited studies on its efficacy for growth promotion and because recommended applications rate have not been clearly established. An understanding of how PA can regulate plant growth, yield and quality of tomato under greenhouse conditions is crucial not only to growers but also to consumers and researchers. In this study, we investigated the biostimulatory effect of varying concentrations of PA for production and increase in nutritional quality of tomato ‘Scotia’ under greenhouse conditions.

## 2. Results

### 2.1. PA Chemical Composition

The chemical composition of PA is presented in Appendix A. The most significant elements were nitrate, nitrite, calcium and potassium. Significant amounts of organic acids (i.e., salicylic acid, oxalic acid, propionic acid, and malic acid) and small amounts of shikimic acid and acylcarnithines were also present.

### 2.2. Morpho-Physiological Response

PA application had no significant (*p* > 0.05) effect on plant height, stem diameter, and the number of branches and flowers (Table 1). Plant height non-significantly increased slightly with low PA concentrations, i.e., 0.25% and 0.5% PA, by *ca*. 5% compared to the control. The highest stem diameter was recorded with 0.25% PA followed by with 2% PA but was not statistically different from that of other treatments. Additionally, plants treated with 0.5% PA increased numbers of branches and flowers by *ca*. 13% and 8%, respectively, compared to that of the control although they were not statistically different (*p* > 0.05). Similarly, PA treatments had no significant (*p* > 0.05) effect on F_v_/F_m_, F_v_/F_o_, and chlorophyll content (Table 2). The effect of PA on F_v_/F_m_ and F_v_/F_o_ was comparable to the control. Likewise, PA had no significant (*p* > 0.05) effect on leaf intracellular CO_2_ and photosynthetic rate (Table 2). However, leaf transpiration rate, sub-stomatal CO_2_, and stomatal conductance were significantly (*p* < 0.001) reduced by PA compared to the control. Plants treated with 0.25% and 0.5% PA showed significant (*p* < 0.001) reductions in these physiological characteristics except for sub-stomatal CO_2_, which was increased by *ca*. 3% with 0.25% PA compared to the control. On the other hand, plants treated with 1% and 2% PA exhibited significant (*p* < 0.001) reductions in leaf transpiration rate, sub-stomatal CO_2_, and stomatal conductance compared to the other PA treatments.

The application of 0.25% PA increased above-ground fresh weight but similar to the control (Figure 1A). However, tomato plants treated with 0.5% and 1% PA reduced the above-ground fresh weight by *ca*. 13% compared to the control. The above-ground dry weight of the tomato plant treated with 0.25% PA was significantly (*p* < 0.005) increased by *ca.* 11% compared to the control (Figure 1B). In contrast, the 0.5% and 1% PA reduced the above-ground plant dry weight but was not significantly (*p* > 0.05) different from those of the control and the 2% PA treatment.

### 2.3. Fruit Yield and Quality

The 0.5% PA treatment increased the total fruit weight by *ca*. 26% although not significantly different from that of 0.25% PA and the control (Figure 2A). However, 2% PA had a significant reduction in total fruit weight, which is not different from that of the 1% PA-treated plants. Similarly, the number of fruits was significantly (*p* < 0.005) increased by *ca*. 66% and *ca*. 34% by 0.5% and 0.25% PA, respectively, compared to the control (Figure 2B). Nevertheless, the application of 2% PA and e control reduced the number of fruits compared to the other PA treatments. Fruit morphological characteristics including polar (Figure 2C) and equatorial diameters (Figure 2D) were not significantly (*p* > 0.05) affected by PA treatment. Tomato fruit juice pH, °Brix, salinity, electric conductivity (EC), total dissolved solids (TSS) and titratable acidity (TA) were significantly (*p* < 0.001) affected by PA treatment (Table 3). Juice pH was significantly (*p* < 0.001) increased by *ca.* 3.3% and 1.3% following the application of 0.25% and 0.5% PA to the plants, respectively, compared to the control. An increase in PA concentration from 1% to 2% did not alter fruit juice pH. The °Brix content of the fruits was increased by *ca.* 13% following the application of 2% PA compared to the control (Table 3). However, °Brix content was significantly (*p* < 0.001) reduced by *ca.* 45% in fruits following the application of 0.25% PA compared to the control.

A significantly (*p* < 0.001) high fruit juice salinity was noticed with the 2% PA treatment compared to the control, while the 0.25% PA recorded the least salinity (Table 3). A considerable increase in fruit electrical conductivity was recorded with the 2% PA, while the least PA of 0.25% reduced fruit juice electrical conductivity. Likewise, the 2% PA recorded the highest fruit juice total dissolved solids (Table 3). Moreover, fruit titratable acidity was significantly (*p* < 0.001) increased by *ca*. 39% upon the application of 2% PA compared to the control (Table 3). Nevertheless, the 0.25% PA had a significant (*p* < 0.001) reduction on fruit TA, which was not different from those of 0.5% PA and 1% PA treatments.

### 2.4. Fruit Biochemicals and Peroxidase Activities

Carotenoid was significantly (*p* < 0.05) increased by the 0.5% PA and 1% PA by *ca*. 20% and 22%, respectively, compared to that of the control (Figure 3A). The carotenoid contents of the 0.5% and 1% PA fruits were not statistically (*p* > 0.05) different from that of the 2% PA, while the carotenoid content of the 0.25% PA fruits was low and comparable to the control. Tomato fruit total phenolics (Figure 3B) and flavonoid were significantly (*p* < 0.001) influenced with PA treatment (Figure 3C). The application of 2% PA exhibited a considerably higher fruit total phenolic compounds (*ca*. 23%) and flavonoid content (*ca*. 39%) compared to the control. The 0.5% PA reduced fruit TPC and flavonoid contents. Total ascorbate was increased by *ca*. 377%, *ca*. 177%, *ca*. 165% and *ca*. 129% following the application of 2%, 0.25%, 1% and 0.5% PA, respectively, compared to the control (Figure 3D). Although 0.5% PA had the highest impact on total fruit protein, it was not statistically (*p* > 0.05) different from those of the 0.25% PA and the control treatments (Figure 3E). However, the 2% PA significantly (*p* < 0.001) reduced total fruit protein content compared to the control. Furthermore, PA caused a significant (*p* < 0.001) reduction in total fruit sugar content (Figure 3F). The 1% PA-treated plants exhibited the least total fruit sugar content, while the 2% PA slightly increased total fruit sugar but was *ca*. 5% lower than that of the control. Furthermore, PA application exhibited a significant (*p* < 0.001) reduction in fruit peroxidase activity (Figure 4). The reduction in peroxidase activity was more obvious in the 0.25% PA fruits followed by the 1% PA and the 2% PA fruits.

### 2.5. Fruit Elemental Composition

Tomato ‘Scotia’ fruit N content was increased by *ca*. 10% upon plant application with 0.25% PA compared to the control but was reduced by the 0.5% PA (Table 4). Fruit Ca was markedly increased by *ca*. 29% upon plant treatment with the 1% PA, but was reduced by the 0.5% PA. Generally, PA had no effect on fruit K compared to the control. However, fruit Mg was increased by *ca*. 13% with the 0.25% PA but was reduced by *ca*. 12% with the 0.5% PA compared to the control. Fruit P content was increased slightly by the 2% PA, which was similar to the effect of the 0.25% PA but was reduced by the 0.5% PA treatment. Fruit Na content increased by *ca*. 59% following the application of 1% PA compared to the control, but was reduced by the 0.5% PA. Variation in PA concentration did not change fruit B content. Overall, Fe, Zn, Mn and Cu, contents were increased with the application of 0.25% PA by *ca*. 8%, *ca*. 8%, *ca*. 9% and *ca*. 15%, respectively, compared to the control. However, the 0.5% PA markedly reduced these four elements in the fruits.

### 2.6. Association between Morpho-Physiological Properties of Tomato Plants and Productivity and Quality in Response to PA Application

Pearson’s correlation coefficient (r) was used to further assess the association amongst the morpho-physiological, yield and quality of tomato plants in response to PA application (Appendix A). The PCA biplot showed a projection of response variables in the factor spaces and explained *ca*. 69% of the total variations in the data set. The results revealed that the number of suckers had a significantly (*p* < 0.05) stronger positive correlation with the number of flowers (r = 0.903) and fruit K content (r = 0.914), while SPAD had a significantly (*p* < 0.05) stronger positive association with leaf intracellular CO_2_ content (r = 0.927) and a negative correlation with photosynthetic rate (r = −0.891). Similarly, leaf transpiration had a significantly strong positively correlated with sub-stomatal CO_2_ content (r = 0.888) and stomatal conductance (r = 0.996) and moderate association with photosynthetic rate (r = 0.608) and total fruit weight (r = 0.651) although this was not statistically significant. Total fruit weight exhibited a significant (*p* < 0.05) and strong positive correlation with plant height (r = 0.943) and fruit number (r = 0.887). However, it had a significantly (*p* < 0.05) strong negative interaction with total phenolics (r = −0.915) and flavonoid content (r = −0.953). Additionally, fruit number has a similar association with plant height (r = 0.915), total phenolics (r = −0.897) and flavonoid content (r = −0.906). Fruit salinity content showed a significantly strong positive correlation with EC (r = 0.998), TDS (r = 0.999) and Brix (r = 0.979), and a negative association with pH (r = −0.864).

## 3. Discussion

Current crop production practices make use of natural products that can boost plant growth and the desirable dietary and nutritional quality without compromising the environment and agroecological systems. Therefore, the functional properties of various natural materials such as PA have recently attracted the interest of farmers and researchers. In this study, although the drench application of PA had no statistically significant effect on tomato ‘Scotia’ plant morphological parameters, they were slightly increased by 0.25% and 0.5% PA concentrations. These results agree with other studies where the foliar application of PA influenced the morphological growth of several plant species including tomato [27], soybean [33], rockmelon [30], and rapeseed [21]. The discovery of karrikins in PA has revolutionized its use in crop production because its signaling and biophysiological activities in plants mimic that of known phytohormones [11,12,15,16]. Moreover, karrikins have been demonstrated to stimulate seed germination and plant growth [12,18]. Hence, the increase in plant growth, although not significant, can be ascribed to the presence of karrikins. Compared to the other elements, N required for vegetative plant growth was considerably high in the PA used for this study. Therefore, the increase in plant growth with PA treatment was reflected in the above-ground fresh and dry weights, which can be attributed to increased nutrient uptake and promotion of cell division and elongation [27].

Stomatal conductance and transpiration rate play a pivotal role in thermoregulation and photosynthesis [34,35]. It was demonstrated that PA and other biostimulants affect stomatal conductance in plants under both stress and non-stress conditions [21,36]. We observed that lower concentrations of PA, i.e., 0.25% or 0.5% PA, exhibited a comparable stomatal conductance and leaf transpiration effect while higher PA concentrations, i.e., >1%, reduced these parameters drastically. A reduction in stomatal conductance is an adaptive strategy used by plants to minimize water loss during water-deficit and other related climatic stress conditions. This scenario adversely affects CO_2_ diffusion and net photosynthesis [37]. Although the photosynthesis rate in the present study was not affected by PA treatment, we surmised that the reduction in stomatal conductance with PA treatment could be due to adaptive thermoregulation of the photosynthesis system and stress mitigation mechanism [35], which will require further investigation.

Plant productivity (i.e., the total number of fruits and yield) increased with PA application as widely reported by many authors [18,21,27,30]. The composition of PA is complex and consists of numerous bioactive compounds including organic acids, phenolics, alcohol, alkane, and ester [18,21]. This suggests that plants with varying genotypic characteristics will respond differently to PA application. In the present study, an increase in the number of tomato fruits and fruit yield were observed with the application of 0.5% PA. The application of 0.5% and 0.25% PA may be considered less toxic to root systems and may promote root growth, thereby enhancing plant nutrient uptake and utilization [25]. Although data on trusses number were not considered, the increase in fruit number in plants treated with lower PA concentrations could suggest that fruit setting was higher in low-PA-treated plants compared to those treated with higher PA concentrations. This was reflected in the correlation analyses where total fruit weight had a strong association with fruit number. From the farmer’s perspective, a slight increase in total fruit yield is considered significant improvement to the overall cashflow. Furthermore, the chemical components of PA might have interacted with and stimulated the activities of various phytohormones including gibberellin, cytokinin, auxin, and various enzymes to enhance plant growth and development as previously reported [21].

Interestingly, determinants of fruit quality such as °Brix, titratable acidity, flavonoid, phenolics, and ascorbate were increased by the 2% PA. This suggests that PA could be used to enhance crop quality for human health and nutritional purposes. These results are inconsistent with the report by Kulkarni et al. [38]. The discrepancies may be due to differences in the tested concentration, time of application, and tomato variety. Generally, tomato fruits are considered an excellent source of phytochemicals including phenolics, flavonoids, and ascorbates, which exhibit high antioxidant properties by scavenging reactive oxygen species (ROS) radicals [2]. Studies demonstrated that higher PA concentration increases the availability of phenolics and organic acids that could affect plant growth [32]. Thus, the increased tomato fruits antioxidants in the present study was highly expected since previous studies have demonstrated that phenolic compounds in PA exhibited high ROS-scavenging activities, reducing power, and anti-lipid peroxidation capacity [8,31].

Accordingly, the present finding may be attributed to the increased phenolics and organic acids as reported in *Citrus limon* [39] and *Olea europaea* [40]. The ROS-scavenging abilities of these phytochemicals protect cells against oxidative stress, which are crucial for preventing chronic diseases including cancers, atherosclerosis, and inflammation disorders [2,3,41]. Moreover, fruit carotenoids are lipophilic pigments essential for human health [42]. Carotenoid content was higher in fruits harvested from plants that were treated with 0.5% and 1% PA compared to the control. This beneficial effect of PA can be attributed to the activation of pathways involved in N metabolism [43]. Furthermore, most plants adapt to stress conditions by accumulating these compounds, which ultimately enhances fruit dietary and nutritional quality. For instance, salinity stress increase TDS, sugar, and antioxidant compounds in tomato fruits [44,45]. Hence, it is plausible that although the 2% PA did not alter the growth of the tomato plants, it stimulated the plants to accumulate these phytochemicals in the fruits.

Mineral elements represent a minute fraction of the fruit dry matter content but constitute a vital component of the quality and nutritional profile of vegetables [46]. The present study demonstrated that the application of 0.25% PA enhanced tomato fruit N, Mg, P, and all the analyzed micronutrients except B. Additionally, the 1% PA increased Ca and Na in the tomato fruits. Some possible explanations could be (1) PA increased the uptake and translocation of mineral elements due to enhanced root growth and root functional activities [24]; (2) PA activated and promoted the expression of transporter genes in root cells for efficient nutrient element transport (not determined); and (3) some bioactive compounds in PA intensified the sink effect resulting in continuous flow and accumulation of these elements [21,47]. Therefore, it can be suggested that the optimal application rate of PA for enhancing tomato fruit elemental composition may range between 0.25% and 1% PA. Similar observations were made following the application of other biostimulants that enhanced the elemental composition of numerous crops including tomato [46,48] and eggplant [49]. Therefore, increased yield and dietary and nutrition quality of tomato can be obtained when the appropriate concentration of PA is applied in a greenhouse production system.

## 4. Materials and Methods

### 4.1. Plant Material and Growing Condition

This research was carried out in the greenhouse located in the Department of Plant, Food, and Environmental Sciences, Faculty of Agriculture, Dalhousie University between November 2020 and February 2021 and repeated in March (spring) and July (summer) 2021. Tomato (*Solanum lycopersicum*) cultivar ‘Scotia’ seeds were purchased from Halifax Seeds (Halifax, Canada). Seeds were sterilized with 10% sodium hypochlorite (NaClO) for 10 min, and thoroughly washed three times with sterile distilled water (ddH_2_O) followed by 70% ethanol sterilization for 5 min, and subsequently washed 5 times with sterile distilled water. The sterilized seeds were germinated in a 32-cell pack containing Pro-Mix^®^ BX (Premier Tech Horticulture, Québec, Canada) and grown for 30 days in a growth chamber with a day/night temperature regime of 25 °C, 16/8 h d^−1^ illumination, 300 μmol m^−2^·s^−1^ light intensity and 70% relative humidity. The seedlings were transplanted at the third to fourth true-leaf stage into 11.35 L-plastic pots containing approximately 1.5 kg of Pro-Mix^®^ BX peat-based soilless medium. The plants were climate hardened for a week before the first treatment application under greenhouse conditions at 28 °C/20 °C (day/night cycle) temperature and 70% relative humidity with a 16 h photoperiod. Supplemental lighting was provided by a 600 W HS2000 high-pressure sodium lamp with NAH600.579 ballast (P.L. Light Systems, Beamsville, Canada) throughout the planting duration.

### 4.2. Experimental Treatment and Design

The five experimental treatments were arranged in a completely randomized design with four replications. The experimental treatments consisted of 0.25%, 0.5%, 1%, and 2% PA, and distilled water was used as a negative control. The PA derived from white pine biomass was obtained from Proton Power Inc. (Lenoir City, TN, USA). The company (Proton Power Inc.) produces and sells graphene and biochar and not PA. The PA is a by-product to them. So, our study, which was funded by the federal agency, was to test this by-product for potential commercialization in the future by which time it will be available to purchase. At present, PA samples may be obtained from Proton Power for only research purposes before it can be available later for purchase. The chemical composition of the PA used in this study is listed in Appendix A. All the treatments were applied biweekly as a soil drench to field capacity, and water-soluble compound fertilizer nitrogen-phosphorus-potassium (20:20:20) was applied at 20-day intervals. Pots were rearranged weekly on the bench to offset unpredictable occurrences due to variations in the environment. The entire study was repeated twice.

### 4.3. Plant Growth and Yield Components

Plant growth parameters were measured at 50 days after transplanting (DAT). Plant height was measured from the stem collar to the highest leaf tip with a ruler and the stem girth (i.e., diameter of the main stem) was measured at 10 cm from the collar with Vernier calipers (Mastercraft^®^, Ontario, Canada). Total numbers of flowers and suckers (i.e., branching) were recorded for each treatment. Intracellular carbon dioxide concentration, net photosynthetic rate, and stomatal conductance were determined from the same four fully expanded leaves per plant using LC*i* portable photosynthesis system (ADC BioScientific Ltd., Hoddesdon, UK). Chlorophyll fluorescence indices including maximum quantum efficiency (F_v_/F_m_) and potential photosynthetic capacity (F_v_/F_o_) were measured on the same leaves using a Chlorophyll fluorometer (Optical Science, Hudson, NH, USA) [50]. Chlorophyll content was measured on the same leaves using a chlorophyll meter (SPAD 502-plus, Spectrum Technologies, Inc., Aurora, IL, USA). The total fresh weight of the above-ground tissues (i.e., leaves and shoot) was measured with a portable balance (Ohaus navigator^®^, ITM Instruments Inc., Sainte-Anne-de-Bellevue, QC, Canada) and subsequently oven-dried at 65 °C for 72 h for dry weight determination. Tomato fruit yield, determined as the total fresh weight of ripe fruits per plant, was recorded using the XT portable balance. The equatorial and polar diameters of the harvested fruits were measured with the digital Vernier caliper.

### 4.4. Fruit Quality and Phytochemical Analysis

At harvest (75DAT), seven representative ripe fruits based on size and color were randomly selected and surface-sterilized with 70% ethanol. The pericarp (containing the epidermis) was excised from the longitudinal part of each fruit using a sterile scalpel blade. The pericarp was immediately frozen in liquid nitrogen and stored in a −80 °C freezer while the remaining fruits were frozen at −20 °C until further analysis. All frozen fruits were thawed at room temperature and fruit total soluble solids (TSS) were determined using a handheld refractometer (Atago, Japan). Briefly, ripe fruits were cut, placed in a clear Ziploc bag and hand squashed. The juice was poured into a 50 mL beaker and 500 µL was used for TSS determination expressed as degree Brix (°Brix). Fruit juice qualities including pH, salinity, total dissolved solids (TDS), and electrical conductivity (EC) were determined with a multi-purpose pH meter (EC 500 ExStik II S/N 252957, EXTECH Instrument, Nashua, New Hampshire, USA). For titratable acidity, 10 mL of juice from each treatment was diluted in 50 mL distilled water, and titratable acidity was determined at an endpoint of pH 8.1 with 0.1 N sodium hydroxide (NaOH). The mean titratable acidity was expressed in citric acid percentage [1]. The elemental composition of the fruits was determined at the Nova Scotia Department of Agriculture Laboratory Services, Truro, using inductively coupled plasma mass spectrometry (PerkinElmer 2100DV, Wellesley, Massachusetts, USA) [51].

#### 4.4.1. Fruit Carotenoid Content

Fruit carotenoid content was determined as described by Lichtenthaler [52]. Briefly, 0.2 g of ground fruit pericarp was homogenized in 2 mL of 80% acetone. The homogenate was centrifuged at 15,000× *g* for 15 min and the absorbance of the supernatant was measured at 646.8, 663.2, and 470 nm using a UV–Vis spectrophotometer with 80% acetone alone as the blank. Total carotenoid content was expressed as µg g^−1^ fresh weight (FW) of the sample.

#### 4.4.2. Total Ascorbate Content

Total ascorbate was measured following the method described by Ma et al. [53] with little modification. Approximately 0.2 g of ground fruit pericarp was homogenized in 1.5 mL ice-cold freshly prepared 5% trichloroacetic acid (TCA). The mixture was vortexed for 2 min and centrifuged at 12,000× *g* for 10 min at 4 °C. A volume of 100 μL of the supernatant was transferred into a new tube and 400 μL phosphate buffer (150 mM potassium dihydrogen phosphate (KH_2_PO_4_) (pH 7.4), 5 mM Ethylenediaminetetraacetic acid (EDTA)) was added. A volume of 100 µL of 10 mM Dithiothreitol (DTT) was added and vortexed for 30 s. A reaction mixture containing 400 µL of 10% (*w*/*v*) trichloroacetic acid (TCA), 400 µL of 44% orthophosphoric acid, 400 µL of 4% (*w*/*v*) α,α-dipyridyl in 70% ethanol and 200 µL of 30 g/L ferric chloride (FeCl_3_) was added to obtain color. The mixture was incubated at 40 °C for 60 min in a shaking incubator and the absorbance was measured at 525 nm. The total ascorbate content was determined using a standard L-ascorbic acid curve and expressed as µmol g^−1^ FW.

#### 4.4.3. Soluble Sugar Content

The total sugar content of the tomato fruits was estimated following the phenol-sulfuric acid method described by Dubois et al. [54]. An amount of 0.2 g of ground fruit pericarp was homogenized in 10 mL of 90% ethanol and the mixture was incubated in a water bath at 60 °C for 60 min. The final volume of the mixture was adjusted to 5 mL with 90% ethanol and centrifuged at 12,000 rpm for 3 min. An aliquot of 1 mL was transferred into a thick-walled glass test tube containing 1 mL of 5% phenol and mixed thoroughly. A volume of 5 mL of concentrated sulfuric acid was added to the reaction mixture, vortexed for 20 s, and incubated in the dark for 15 min. The mixture was cooled to room temperature and the absorbance was measured at 490 nm against a blank. Total sugar was calculated using a standard sugar curve and expressed as µg of glucose g^−1^ FW.

#### 4.4.4. Total Phenolics Content

Total phenolics content (TPC) was determined by the Folin–Ciocalteu assay described by Ainsworth and Gillespie [55] with little modification. An amount of 0.2 g of ground fruit pericarp was homogenized in 1.5 mL of ice-cold 95% methanol and incubated in the dark at room temperature for 48 h. The mixture was centrifuged at 13,000× *g* for 5 min before mixing 100 µL of the supernatant to 200 µL of 10% (*v*/*v*) Folin–Ciocalteau reagent. The mixture was vortexed for 5 min, mixed with 800 µL 700 mM Na_2_CO_3_, and incubated in the dark at room temperature for 2 h. The absorbance of the supernatant was measured at 765 nm against a blank. TPC was calculated using a gallic acid standard curve and expressed as mg gallic acid equivalents per g FW (mg GAE g^−1^ FW).

#### 4.4.5. Total Flavonoid Content

Total flavonoid was estimated following the colorimetric method described by Chang et al. [56]. An amount of 0.2 g of ground fruit pericarp was homogenized in 1.5 mL of ice-cold 95% methanol followed by centrifugation at 15,000× *g* for 10 min. A volume of 500 µL of supernatant was added to a reaction mixture containing 1.5 mL of 95% methanol, 0.1 mL of 10% aluminum chloride (AlCl_3_), 0.1 mL of 1 M potassium acetate, and 2.8 mL distilled water. The mixture was incubated at room temperature for 30 min and the absorbance was measured at 415 nm against a blank lacking AlCl_3_. Total flavonoid content was estimated using quercetin equivalents and expressed as percentage flavonoid using the formula:$$\mathrm{Total}\;\mathrm{flavonoid}=\;\frac{\left(\left[\mathrm{flavonoids}\right]\left(µg/\mathrm{mL}\right)\times \;\mathrm{total}\;\mathrm{volume}\;\mathrm{of}\;\mathrm{methanolic}\;\mathrm{extract}\;\left(\mathrm{mL}\right)\right)\;}{\mathrm{mass}\;\mathrm{of}\;\mathrm{extract}\;\left(g\right)\;}$$

#### 4.4.6. Protein Content and Peroxidase Activity

For protein content and antioxidant enzyme activity, approximately 0.2 g of ground sample was homogenized in 3 mL ice-cold extraction buffer (50 mM potassium phosphate buffer (pH 7.0), 1% polyvinylpyrrolidone, and 0.1 mM EDTA). The homogenate was centrifuged at 15,000× *g* for 20 min at 4 °C. The supernatant (crude enzyme extract) was transferred to a new microfuge tube on ice and the protein content was measured at 595 nm after 5 min of mixing with Bradford’s reagent [57]. The protein content was estimated from a standard curve of bovine serum albumin (200–900 µg mL^−1^). Peroxidase (POD, EC 1.11.1.7) activity was determined using Pyrogallol as substrate according to Chance and Maehly [58] with little modification. The reaction mixture consisted of 100 mM potassium-phosphate buffer (pH 6.0), 5% pyrogallol, 0.5 % H_2_O_2_ and 100 μL of crude enzyme extract. Following reaction mixture incubation at 25 °C for 5 min, 1 mL of 2.5 N H_2_SO_4_ was added to stop the reaction and the absorbance was read at 420 nm against a blank (ddH_2_O). One unit of POD forms 1 mg of purpurogallin from pyrogallol in 20 s at pH 6.0 at 20 °C.

### 4.5. Statistical Analysis

All data obtained were subjected to one-way analysis of variance (ANOVA) with the averages of the two experiments using Minitab statistical software version 20 (Minitab Inc., State College, PA, USA). Treatment means were compared using Fisher’s least significant difference (LSD) post hoc test at *p* ≤ 0.05. Pearson’s correlation analysis was performed using XLSTAT version 19.1 (Addinsoft, New York, NY, USA).

## 5. Conclusions

In conclusion, the drench application of low PA concentrations of 0.25% and 0.5% increases the morpho-physiological response of tomato plants. Overall, the application of 0.5% PA enhances the number of fruits and yield of tomato but reduces the quality of the fruits. Alternatively, the application of 0.25% PA will increase the elemental composition of tomato fruits. Additionally, the drench application of 2% PA can be considered stressful to tomato plants, but significantly enhanced fruit phytochemical contents including total phenolics and flavonoids and can be adopted to improve the nutritional and health benefits of tomato fruits. Hence, PA represents a novel natural product for improvement of plant growth, productivity, and nutritional content of tomato and other plants. However, further investigation is required to elucidate the molecular basis of the effect of PA on different plant species.

## Figures and Tables

**Figure 1 plants-11-01650-f001:**
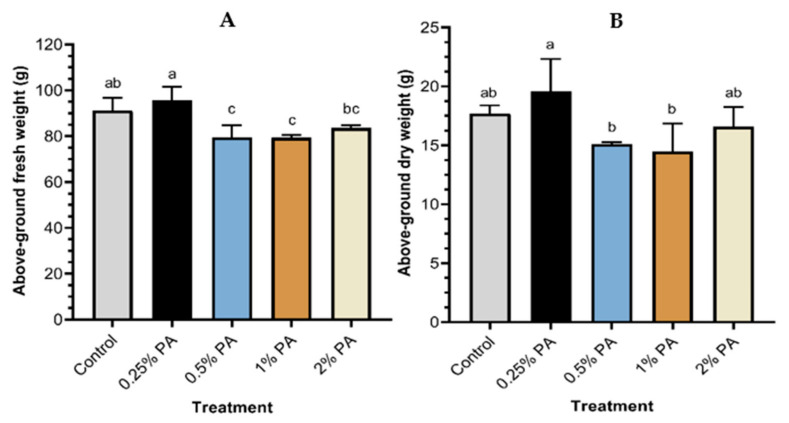
Pyroligneous acid effect on tomato plant above-ground biomass: (**A**) fresh weight and (**B**) dry weight. Values are the means of four replicates and different lowercase alphabetical letters indicate significant (*p* < 0.05) difference according to Fisher’s least significant difference (LSD) post hoc test. Error bars show the standard deviations.

**Figure 2 plants-11-01650-f002:**
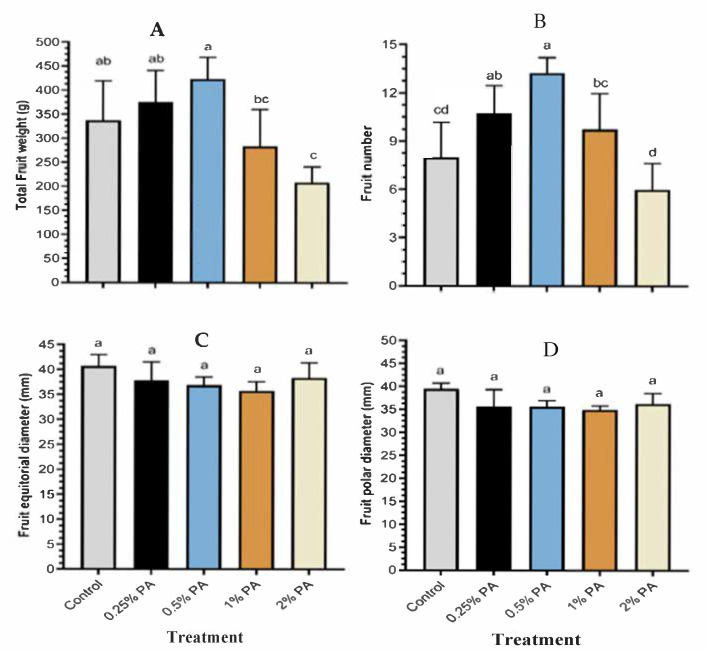
Fruit yield of tomato ‘Scotia’ in response to pyroligneous acid treatment: (**A**) total fruit weight, (**B**) fruit number, (**C**) fruit polar diameter, and (**D**) fruit equatorial diameter. Values are the means of four replicates and different lowercase alphabetical letters indicate significant (*p* < 0.05) difference according to Fisher’s least significant difference (LSD) post hoc test. Error bars show the standard deviations.

**Figure 3 plants-11-01650-f003:**
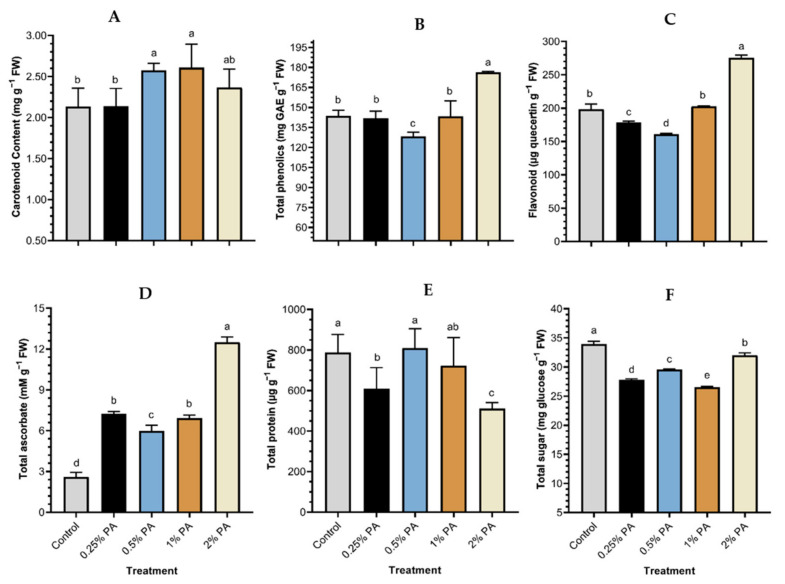
Tomato ‘Scotia’ fruit biochemical content in response to pyroligneous acid treatment: (**A**) carotenoid content, (**B**) total phenolic content, (**C**) flavonoid content, (**D**) total ascorbate content, (**E**) total protein content, and (**F**) total sugar content. Values are the means of four replicates and different lowercase alphabetical letters indicate significant (*p* < 0.05) difference according to Fisher’s least significant difference (LSD) post hoc test. Error bars show the standard deviations.

**Figure 4 plants-11-01650-f004:**
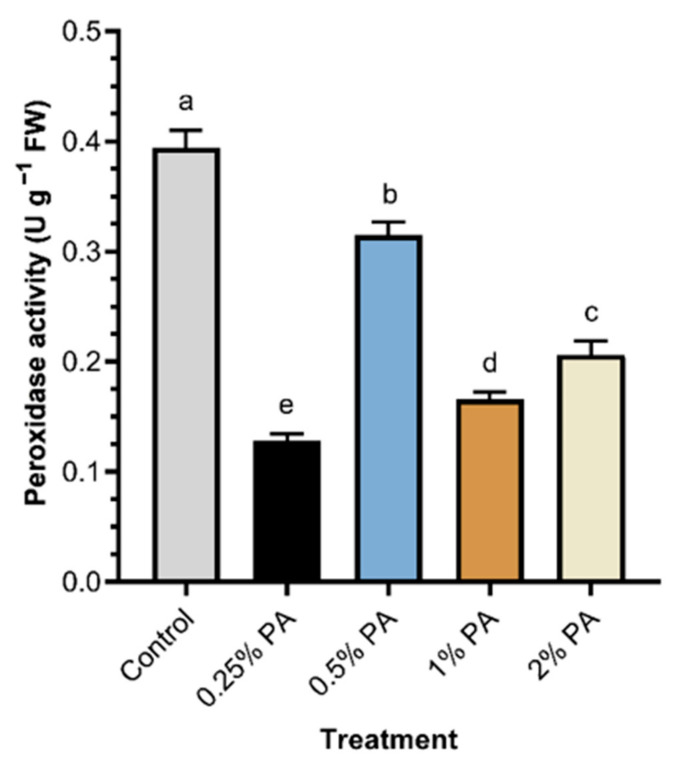
Peroxidase activity of tomato ‘Scotia’ fruit in response to pyroligneous acid treatment. Values are the means of four replicates and different lowercase alphabetical letters indicate significant (*p* < 0.05) difference according to Fisher’s least significant difference (LSD) post hoc test. Error bars show the standard deviations.

**Table 1 plants-11-01650-t001:** Morphological response of tomato ‘Scotia’ plants treated with pyroligneous acid (PA).

Treatment	Plant Height (cm)	Stem Diameter (mm)	Branch Number	Flower Number
Control	57.82 ± 2.86 a	9.60 ± 0.80 a	6.52 ± 0.58 a	33.50 ± 7.23 a
0.25% PA	60.50 ± 5.79 a	10.02 ± 0.61 a	5.81 ± 0.96 a	27.25 ± 7.80 a
0.5% PA	60.62 ± 5.23 a	9.32 ± 0.77 a	7.04 ± 0.82 a	38.00 ± 8.41 a
1% PA	57.71 ± 6.40 a	9.51 ± 0.92 a	5.70 ± 1.73 a	31.00 ± 11.86 a
2% PA	56.07 ± 2.97 a	9.81 ± 0.49 a	6.38 ± 1.50 a	32.25 ± 11.41 a
*p*-value	0.565	0.689	0.480	0.622

Values are the means ± SD of four replicates and different letters indicate significant (*p* < 0.05) difference according to Fisher’s least significant difference (LSD) post hoc test.

**Table 2 plants-11-01650-t002:** Physiological response of tomato ‘Scotia’ plants treated with pyroligneous acid (PA).

Treatment	F_v_/F_o_	F_v_/F_m_	SPAD	Intra Cellular CO_2_ (µmol mol^−1^)	A (μmol m^−2^ s^−1^)	E (mol m^−2^ s^−1^)	Ci (µmol mol^−1^)	g_s_ (mol m^−2^ s^−1^)
Control	4.16 ± 0.41 a	0.80 ± 0.01 a	34.14 ± 5.80 a	410.56 ± 6.13 a	2.15 ± 0.60 a	2.53 ± 0.52 a	360.70 ± 30.46 ab	0.11 ± 0.02 a
0.25% PA	4.06 ± 0.27 a	0.81 ± 0.01 a	36.59 ± 3.74 a	417.74 ± 8.72 a	1.80 ± 0.84 a	2.16 ± 0.60 ab	370.27 ± 19.04 a	0.09 ± 0.01 ab
0.5% PA	3.96 ± 0.33 a	0.80 ± 0.01 a	34.07 ± 2.96 a	410.85 ± 6.61 a	2.19 ± 0.80 a	1.95 ± 0.71 b	343.01 ± 35.68 b c	0.08 ± 0.03 b
1% PA	4.07 ± 0.34 a	0.80 ± 0.01 a	35.57 ± 5.14 a	413.55 ± 13.84 a	1.80 ± 0.79 a	1.28 ± 1.03 c	325.23 ± 42.80 c	0.05 ± 0.04 c
2% PA	3.91 ± 0.24 a	0.80 ± 0.01 a	37.13 ± 6.32 a	415.68 ± 14.65 a	1.84 ± 1.38 a	1.35 ± 0.59 c	332.50 ± 41.80 c	0.05 ± 0.04 c
*p*-value	0.196	0.188	0.262	0.226	0.534	<0.001	<0.001	<0.001

A: photosynthetic rate; E: transpiration rate; g_s_: stomatal conductance; Ci: sub-stomatal CO_2_. Values are the means ± SD of four replicates and different letters indicate significant (*p* < 0.05) difference according to Fisher’s least significant difference (LSD) post hoc test.

**Table 3 plants-11-01650-t003:** Chemical quality of tomato ‘Scotia’ fruits from plants treated with pyroligneous acid (PA).

Treatment	Juice pH	°Brix	Salinity (g L^−1^)	EC (mS)	TDS (g L^−1^)	TA (% Citric Acid)
Control	3.60 ± 0.04 c	5.67 ± 0.05 b	2.95 ± 0.02 b	5.42 ± 0.03 b	3.80 ± 0.01 b	0.26 ± 0.01 b
0.25% PA	3.72 ± 0.01 a	3.12 ± 0.05 d	1.66 ± 0.02 e	3.11 ± 0.08 e	2.20 ± 0.02 e	0.23 ± 0.03 c
0.5% PA	3.67 ± 0.01 b	5.62 ± 0.13 b	2.68 ± 0.03 c	5.01 ± 0.07 c	3.44 ± 0.07 c	0.24 ± 0.01 b c
1% PA	3.62 ± 0.03 c	5.20 ± 0.14 c	2.50 ± 0.03 d	4.65 ± 0.04 d	3.22 ± 0.02 d	0.23 ± 0.01 b c
2% PA	3.62 ± 0.03 c	6.42 ± 0.10 a	3.01 ± 0.03 a	5.71 ± 0.04 a	3.94 ± 0.04 a	0.36 ± 0.01 a
*p*-value	<0.001	<0.001	<0.001	<0.001	<0.001	<0.001

EC: electrical conductivity; TDS: total dissolved solids; TA: titratable acidity. Values are the means ± SD of four replicates and different letters indicate significant (*p* < 0.05) difference according to Fisher’s least significant difference (LSD) post hoc test.

**Table 4 plants-11-01650-t004:** Tomato ‘Scotia’ fruit elemental composition in response to pyroligneous acid (PA) treatments.

Element	Treatment						
	Control	0.25% PA	0.5% PA	1% PA	2% PA	Mean	CV (%)
Nitrogen (N %)	1.68	1.84	1.44	1.61	1.56	1.63	9.12
Calcium (Ca %)	0.24	0.29	0.23	0.31	0.28	0.27	12.39
Potassium (K %)	2.68	2.27	2.67	2.32	2.59	2.51	7.91
Magnesium (Mg %)	0.17	0.19	0.15	0.17	0.17	0.17	8.21
Phosphorus (P %)	0.44	0.46	0.41	0.42	0.47	0.44	5.60
Sodium (Na %)	0.02	0.03	0.02	0.04	0.02	0.02	26.28
Boron (B mg L^−1^)	12.61	13.61	12.59	13.91	13.62	13.27	4.69
Copper (Cu mg L^−1^)	7.51	8.86	5.98	6.53	7.02	7.18	15.29
Iron (Fe mg L^−1^)	42.46	49.87	40.08	43.37	45.00	44.16	8.28
Manganese (Mn mg L^−1^)	26.14	28.37	22.58	27.49	25.06	25.93	8.71
Zinc (Zn mg L^−1^)	14.81	17.71	14.80	14.68	16.25	15.65	8.44

CV = coefficient of variation.

## Data Availability

Not applicable.

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
