# Peer review of "Effect of Pyroligneous Acid on the Productivity and Nutritional Quality of Greenhouse Tomato"

_plants, 2022, doi:10.3390/plants11131650_

Round 1
Reviewer 1 Report
The authors nicely revised the manuscript. However, it is always advised to mark the changes with a different color.
Still, the paper needs a careful proof check and I also suggest using data standard units and symbols.
e.g. Fv/Fo, gs, g L-1, GAE g-1 FW, etc.
Also, be consistent in using such units and symbols.
The results are poorly written. The authors are just focused on the significance as expressed by the probability. However, I suggest mentioning the extent of changes, e.g. % increase/ decrease over control or like this.
In the graphs, omit the decimal point of Y-axis values unless it is less than one.
Author Response
Dear Reviewer:
Thanks for your comments/suggestions which were all taken into consideration to improve the manuscript.

Reviewer 2 Report
Authors addressed the comments and suggestions, and provided corrections and/or clarifications to the manuscript. Figures and tables were also corrected as suggested. The manuscript reads well and presents a more clear analysis of the effect of PA on Tomato plant.
Author Response
Thanks for your previous comments/suggestions which helped to improve the manuscript.
Reviewer 3 Report
The manuscript could be accepted in the present from
Author Response
Thanks for your previous comments/suggestions which helped to improve the manuscript.
This manuscript is a resubmission of an earlier submission. The following is a list of the peer review reports and author responses from that submission.
Round 1
Reviewer 1 Report
This paper presents the role of pyroligneous acid in improving the yield and quality of tomato. The positive effect of PA in improving plant performance has not been studied extensively. Therefore, this research has some novelty.
However, this research is mostly focused on the effect of PA but their mechanism is not revealed. How PA is being uptaken and how they regulate other metabolites are not clearly investigated.
The authors have measured a bunch of parameters but these are not directly influenced by PA. For example, the antioxidant defense may be regulated by any exogenous protectants.
The data are inconsistent. For example, some data such as FW and DW do not maintain a dose-dependent manner. So, which dose of PA is the best for tomato yield and quality is not conclusive.
I am wondering why control is placed at the end! It must be placed first.
Correlation and PCA are not feasible for such experiments.
The paper is too wordly. Please concise results and discussion.
References are not properly formatted.
Reviewer 2 Report
This paper is well written, authors provide justification for the experimental variables they have chosen to study, and they examine multiple plant characteristics during different stages of development. Various analytical methods are used to evaluate plant development, as well as changes in nutritional profile of fruit with addition of PA at different concentrations. This is an interesting and valuable research, which demonstrates that drench application of PA can potentially be of value in agriculture with respect to improving growth and nutritional characteristics of tomato.
Here is a list of comments and suggestions for authors to consider before publication:
- Experimental design section states that there were four replicates collected, and then on line 130 authors state that the entire study was repeated twice. Some figures in the results section specify that the data is based on four replicates, however I did not see anywhere if the authors used samples from the repeated study, if it was indeed conducted. If the entire study was repeated twice, there should be 8 replicates for each experiment, and then authors must specify why only four replicates were used and how/why these four were selected.
- P value in some tables (2 and 3) are reported as 0.000, however it would be more correct to either present all p-values in scientific notation, or report particularly low values as <0.001.
- Line 243 –a typo, based on Table 1 the values should be 0.25% and 0.5% PA.
- It could be more effective to place control sample data above (tables)/to the left of (figures) the other PA values, as to create a more logical flow from the lowest to the highest concentration of PA tested.
- It would be interesting to see the effect of PA on taste characteristics in the future studies. While more desirable nutritionally, an increase in acidity and flavonoid content, along with reduced sugar, might lead to less acceptable taste.
Reviewer 3 Report
In this manuscript Ofoe et al. reports investigated the biostimulatory effect of varying concentrations of PA application on tomato ) plant growth and fruit quality under greenhouse conditions. The experimental design is appropriate, the data is well presented and the manuscript is very well written.
However, some minor changes are needed, before being accepted for publication:
1) In the introduction (line 57) the authors write: "The chemical composition of PA mainly depends on the temperature, heating rate, feedstock, and residence time ". However, they did non report the chemical composition of the selected PA in the Method section. Moreover, in the Conclusion, they state that "However, further investigation is required to elucidate the molecular basis of PA effect on different". Thus, adding information on the chemical composition of the PA could be useful to compare future results, with those presented in this manuscript. Thus, any information about the chemical composition of the PA would increase the quality of the present work.
2) in the section "2.5. Statistical analysis" the authors describe only the ANOVA and LSD. However, they also performed 2D-PCA and Pearson Correlation. Have they used the same software (Minitab) also for this further analysis?
3) Figure 5 is not easy-to-read for readers with no strong statistical background or knowledge. Could the authors modify Figure 5 adding the Score Plot and the Loading Plot?
Round 2
Reviewer 1 Report
I still do not believe that such PCA is important and, therefor, this paper should be consixdered for publication.
The paper is still too wordy.